# Behavioral determinants for glycemic control among type 2 diabetic patients in Hosanna town; institution based unmatched case control study

**Belay Bancha** (ORCID)*, **Begidu Bashe, Hana Seyfu, Abraham Samuel**

Hosanna Health Science College, Hosanna, Ethiopia

* banchabelay@gmail.com

## Abstract

### Backgrounds

Together with other non-communicable diseases, diabetes mellitus (DM) remains main health threats to human beings throughout the world. Despite an alarming increment of its trend in Ethiopia, majority of diabetic patients failed to attain glycemic control. Thus, this study aimed at identifying modifiable lifestyles for glycemic control among patients with type 2 diabetic mellitus.

### Objectives

To assess behavioral determinants of glycemic control among type 2 diabetic patients attending Wachemo University Nigist Eleni Mohammed Memorial Teaching Hospital (WUNEMMTH) in Hosanna, Ethiopia.

### Methods

Institution based unmatched case control study was conducted from May15-July 15, 2023. The questionnaire was digitalized to open data kit (ODK) for android version. Data was collected by simple random sampling technique and analyzed using SPSS version 23. Descriptive and logistic regression analysis were performed. Adjusted odds ratios and 95% confidence intervals were used to measure the effect size. Level of statistical significance was declared at a p-value of $< 0.05$.

### Results

From 232 expectants, 226 (cases = 113 and controls = 113) were participated, making the response rate 97.41%. After adjusting for others, non-adherence to fruit and vegetable intake (AOR = 3.38, 95% CI = 1.73-6.60), non-adherence to physical exercise (AOR = 4.94, 95% CI = 2.38-10.27), poor diabetic self-efficacy (AOR = 5.51, 95% CI = 2.85-10.66) and high body mass index (AOR = 3.68, 95% CI = 1.73-7.82) were independent determinants for poor glycemic control among T2DM clients.

**Data availability statement:** All relevant data are within the paper and its Supporting Information files.

**Funding:** The author(s) received no specific funding for this work.

**Competing interests:** The authors have declared that no competing interests exist.

**Abbreviations:** AOR, Adjusted Odds Ratio; BMI, Body Mass Index; DM, Diabetes Mellitus; RBG, Random Blood Glucose; T2DM, Type 2 Diabetes Mellitus; WUNEMMTH, Wachamo University Nigist Eleni Mohammed Memorial Teaching Hospital.

## Conclusion and recommendations

Self-efficacy and healthy lifestyle were behavioral determinants for glycemic control among T2DM patients. Thus, interventions targeting modifiable lifestyle should be an integral part of the management along with standard pharmacotherapy.

## Introduction

Diabetes mellitus (DM) is a chronic metabolic disorders characterized by hyperglycemia. It is one of the largest global concerns, imposing heavy burden on public health and socio-economic development across the world [1–4]. Together with cardiovascular diseases, cancer, stroke and respiratory disease, DM appears to be the main health threats and it is one of the five priority non-communicable diseases (NCDs) in the action plan of United Nations (UN) and World Health Organization (WHO) [5,6].

Uncontrolled DM considerably affects the individuals' quality of life, social interaction, and economic productivity. In 2017, approximately 462 million individuals, corresponding to 6.28% of the world's population, were affected by type 2 diabetes mellitus (T2DM); costing US$ 850 billion health care expenditure for its management. The global prevalence of T2DM is projected to rise to 7079 individuals per 100,000 by 2030, reflecting an exponential rise to epidemic proportions across the world [1,6–9].

About 79% of adults with diabetes reside in low and middle- income countries, where the burden is very significant. International Diabetes Federation (IDF) estimates of 2021 shown that 1 in 22 adults with DM live in Africa. In the region, its prevalence among adults between 20–79 years was 16 million in 2017 and projected to be 41 million in 2045 [1,2,9,10]. The second most populated African country, Ethiopia, is also one of these countries where the trend of diabetes mellitus burden is alarmingly increasing [1,8]. However, 59.2–73.8% of diabetic patients in Ethiopia failed to attain the recommended glycemic control [11–14].

Along with socio-demographic and epidemiological transitions, modifiable risky lifestyles such as unhealthy dietary patterns, lack of physical exercise, excess body weight, sedentary lifestyle and tobacco and alcohol consumption are highly responsible for the current increasing incidence and prevalence of T2DM [1,7,8,15]. Managing patients with diabetes involve complex, long-lasting, and costly endeavors [5,6]. However, diabetic self-efficacy, adherence to healthy lifestyles and self-care activities such as self-monitoring of blood glucose, self-monitoring of complications, adherence to follow-up appointment and medication for tight glycemic control can delay or prevent the incidence and progression of complications associated with T2DM (S1 Fig). Targeting such behavioral factors are the most feasible, efficient and effective public health measure. Of the lifestyle modifications, adherence to dietary and physical activity recommendations help to maintain optimal body weight that subsequently combat glucose intolerance [8,15–17]. However, previous studies in Ethiopia not well addressed modifiable risk factors for glycemic control. Additionally, the effect of self-efficacy for glycemic control was not studied. Thus, this study has assessed the behavioral determinants of glycemic control among T2DM patients under follow-up care.

## Conceptual framework

A conceptual framework was developed to assess behavioral determinants of glycemic control among patients with T2DM (S1 Fig).

## Objective

To assess behavioral determinants of glycemic control among type 2 diabetic patients attending Wachemo University Nigist Eleni Mohammed Memorial Teaching Hospital (WUNEM-MTH) in Hosanna town.

## Materials and methods

### Study design, area and period

An institutional based unmatched case control study design was conducted from May 15-July 15, 2023 in WUNEMMTH, Hosanna town, Central Ethiopia Regional State. The town is located 232 Km Southwest of national Capital, Addis Ababa. According to 2007 Central Statistical Agency report [18], estimated population living in the town was projected to 188,192; of which 91,535 are males and 96,657 are females with 5.7% prevalence of T2DM patients [19].

### Sample size and sampling technique

Sample size was computed using Epi Info version 7, with the assumptions of 95% significance level, 80% power, case to control ratio of 1:1, odds ratio of 2.42 [12], 57.52% of controls exposed (Table 1), 5% margin of error, and 10% non-response rate. Based on their mean FBG level of three consecutive visits, cases and controls were identified from May 1-5, 2023. After obtaining sampling frame from client registry, simple random sampling (SRS) technique was used to identify study participants.

### Population

All T2DM patients living in Hosanna town were source population whereas all clients with T2DM who were registered at WUNEMMTH for T2DM follow-up visit were study population. The sampling unit consisted of randomly selected T2DM patients from whom data was collected.

### Identification of cases and controls

Cases and controls were identified from patient record; controls were patients with good glycemic control whose previous three months' mean fasting blood glucose (FBG) level was < 130 mg/dl, whereas clients were identified as cases when FBG was ≥ 130mg/dl [21].

### Inclusion and exclusion criteria

Patients with T2DM and having at least three months consecutive follow-up in diabetes clinic in the Hospital were included in both cases and controls. Patients with T2DM whose age less than 18 years, mentally unstable or critically ill at time of data collection were excluded.

**Table 1. Sample size for behavioral determinants of glycemic control among T2DM patients in Hosanna town, Central Ethiopia, 2024.**

| Variable | Power | % cases with exposure | % control with exposure | AOR | Estimated sample size | References |
|---|---|---|---|---|---|---|
| Self-monitoring of plasma glucose | 80 | 20 | 29.8 | 3.44 | 100 | [20] |
| Physical activity | 80 | 21.5 | 44.9 | 4.79 | 70 | |
| Alcohol intake | 80 | 38.4 | 15.9 | 3.3 | 138 | [19] |
| Adhere to follow up | 80 | 76.6 | 57.5 | 2.42 | 210 | [12] |

With the above assumptions, the final sample size was 232, i.e., 116 cases and 116 controls.

## Data collection tools and procedures

Questionnaire was adapted from Ethiopian Demographic and Health Survey 2016 [22] and WHO stepwise for chronic diseases [23] and relevant literature [4,19–21,24,25]. Diabetic self-efficacy was determined using 15 items with five point Likert scale [26]. The questionnaire was developed in English and then it was translated to Amharic, then back translated to English to confirm consistency. English version questionnaire was digitalized to open data kit (ODK) for android version. Socio-demographic characteristics, self-care activities, adherence to healthy lifestyles and medication and diabetic self-efficacy data were collected through interviewing patients. Patients' weight and height were measured at time of interview and FBG was obtained from patients' medical record (S2 File).

## Operational definitions

- **Poor glycemic control:** when a patient is under regular follow-up but FBG is ≥ 130 mg/dl.

- **Good glycemic control:** when a patient is under regular follow-up and FBG is < 130 mg/dl

- **Adherence to medication:** if a patient took all his/her anti-diabetic medication in the last seven days [20].

- **Healthy diet:** is comprised of fruits and vegetables and foods high in fiber and whole grain, but low in carbohydrates (non-starchy), fats (milk and dairy products), and sugars.

- **Adherence to dietary recommendations:** If a respondent follows healthy diet for glycemic control for more than 3 days in the last seven days.

- **Adherence to exercise:** If a respondent follows the recommended level of exercise for more than 3 days in the last seven days.

- **Alcohol consumption:** reported consumption of any type and amount of alcohol one week prior to the survey.

- **Diabetic self-efficacy:** was measured using 15 items with 5-point Likert scale of diabetic self-efficacy; below mean score was used as cutoff to declare poor self-efficacy.

## Study variables

Dependent variable was glycemic control where as independent variables were self-efficacy to diabetic control, adherence to dietary recommendations, adherence to Physical exercises, body weight regulation, adherence to follow-up appointment, adherence to medication, self-monitor of blood glucose level and self-monitor of complications (S1 Fig and S2 File).

## Data quality management

Five data collectors and two supervisors received two days training to ensure the quality of the data. Pre-test was done on 5% of sample size in a setting similar to the study area. Alpha coefficient was computed to assess the internal consistency; tool modification was considered based on pre-test result. All collected data were submitted to principal investigators who checked for consistency and completeness; inconsistences were discussed on daily bases.

## Defining dietary pattern

Principal component analysis (PCA) was used to assess dietary behavior based on five dietary items. First, all the study participants were asked about the frequency of consumption of

healthy diets in a typical week; those who consumed these dietary items for more than three days were assigned a score "1" and those who consumed them for less than or equal to three days were assigned a score "2". All five dietary items were retained in two component factor that explains a total variance of 84.75. Three items (low glycemic index diet, High fiber diet, and evenly spacing of carbohydrate) loaded in component one; was given a name "low glycemic high fiber diet". Fruits and vegetables loaded in component two and was assigned a name "fruit and vegetable". Using the median as a cutoff, each dietary pattern was ranked into two; good and poor adherence to dietary recommendations.

## Data analysis procedures

ODK was used for data collection and then exported to Statistical Package for Social science (SPSS) version 23 for analysis. Missed values and normality for continuous variables were checked, recoded and descriptive analysis was performed. Glycemic control was labelled based on the predefined case and control definition to fit for logistic regression model. First, glycemic control was regressed on each independent variable to identify potential candidate variables for multivariable logistic regression. The Variables identified with p-value of $< 0.25$ in bivariate analysis were then entered into multivariate analysis using backward removal method to account for potential confounding effects of each other. After checking assumptions for the model, multivariable logistic regression analysis was fitted to identify behavioral determinants of glycemic control among T2DM patients. Finally, model fitness was checked by Hosmer-Lemeshow goodness-of fit test and the model was a good fit. Adjusted odds ratio (AOR) and corresponding 95% CI were used in measuring effect size. Level of statistical significance was declared at a p-value of $< 0.05$. Data set used for this analysis is available as a supporting information (S3 Data Set).

## Ethics approval and consent to participate

The study was approved by the Institutional Review Board of Hosanna Health Science College; with the reference number 4720. All study participants were briefed about purpose of the study, participants' recruitment, potential benefits and risks in taking part in this study. All participants were informed that they had the right to refuse or decline from the study at any point. Before data collection, written informed consent was obtained. Authors had no access to information that could identify individual participants during or after data collection. The information provided by each respondent was kept confidential adhering to research ethics.

## Results

### Socio-demographic characteristics

From a total of 232 study units, 226 (113 cases and 113controls) were participated in the study, making a response rate 97.4%. Of these, 130 (57.5%) (cases = 60 and controls = 70) were male, over one third (34.5%) were in age category of 40-49 years. More than two third (68.6%) and three fourth (75.7%) of the study participants were Protestant religion follower and Hadiya by ethnicity respectively. Additionally, 81.4% of the participants attended various level of formal education and 87.6% were in marital union (Table 2).

### Diabetic self-care behaviors

In a typical week, 48 (21.2%) cases and 81 (35.8%) controls consumed fruits for more than three times. Additionally, 61 (27%) cases and 88 (38.9%) controls were adherent to recommended vegetable consumption. In this study, 36.3% cases and 35.4% of controls were

**Table 2. Socio-demographic characteristics of type 2 diabetic patients attending chronic OPD clinic in WUNEM-MTH, Hosanna, Central Ethiopia, 2024 (n = 226).**

| Variables | Category | Frequency (%) |
|---|---|---|
| Sex | Male | 130(57.5) |
| | Female | 96 (42.5) |
| Age | 20–29 years | 33(14.6) |
| | 30–39 years | 46 (20.4) |
| | 40–49 years | 78 (34.5) |
| | 50–59 years | 34 (15.0) |
| | >=60 years | 35 (15.5) |
| Religion | Orthodox | 45 (19.9) |
| | Protestant | 155 (68.6) |
| | Muslim | 18 (8.0) |
| | Catholic | 8 (3.5) |
| Level of educational | No formal education | 42 (18.6) |
| | Primary (1–8) | 92 (40.7) |
| | Secondary (9–12) | 56 (24.8) |
| | Higher Education | 36 (15.9) |
| Residence | Urban | 141 (62.4) |
| | Rural | 85 (37.6) |
| Occupation | Government employee | 37 (16.4) |
| | Non-governmental Organization employee | 11 (4.9) |
| | Private employed | 31 (13.7) |
| | Merchant | 44 (19.5) |
| | Housewife | 64 (28.3) |
| | Farmer | 25 (11.1) |
| | Others | 14 (6.2) |
| Marital status | Single | 22 (9.7) |
| | Married | 198 (87.6) |
| | Widowed | 4 (1.8) |
| | Divorced | 2 (0.9) |
| Ethnicity | Hadiya | 171 (75.7) |
| | Kambata | 19 (8.4) |
| | Amhara | 14 (6.2) |
| | Silte | 10 (4.4) |
| | Gurage | 8 (3.5) |

adherent to recommended low glycemic and high fiber dietary intake in a typical week. After PCA, two dietary patterns were created. Based on this analysis, 45.2% (31% cases and 14.2% controls) and 50.9% (33.6% cases 17.3% controls) T2DM patients were non-adherent to fruit and vegetable and high fiber low glycemic index diet recommendation respectively (S4 Fig).

This study revealed that 11.1% (5.3 cases and 5.8% controls) of respondents consumed some type of alcoholic drink within seven days before the interview and 14.5% of respondents (9.3% cases and 6.2% controls) had history of cigarette smoking. About two third (40.7% cases and 24.8% controls) of study participants were not adherent to physical exercise recommendations.

Less than two in ten (7.5% cases and 10.6% controls) respondents had access to a device for self-monitoring of their blood glucose, 24.8% of respondents (18.1% cases and 6.6% controls) were non-compliant to prescribed anti-diabetic medications in last seven days and 28.8%

(19.9% cases and 8.8% controls) had high body mass index (BMI). Only 13.3% cases and 36.3% controls had good diabetic self-efficacy.

## Determinants of glycemic control

In Bi-variable analysis, status of adherence to fruit and vegetable intake, adherence to fiber intake, physical exercise, cigarette smoking, compliance to anti-diabetic medications, diabetic self-efficacy and BMI were determinants of glycemic control among T2DM patients at p-value of < 0.25.

After adjusting for variables significant in bivariate analysis, adherence to fruit and vegetable intake, adherence to physical exercise, diabetic self-efficacy and BMI were independent determinants of glycemic control among T2DM patients. The odds of poor glycemic control was 3.38 times (AOR = 3.38, 95% CI = 1.73-6.60) and 4.94 times (AOR = 4.94, 95% CI = 2.38-10.27) more likely among patients who were not adherent to fruit and vegetable intake and physical exercise recommendations respectively. The likelihood of poor glycemic control was 5.51 times (AOR = 5.51, 95% CI = 2.85-10.66) among T2DM patients with poor diabetic self-efficacy. Additionally, the risk of poor glycemic control was 3.68 times (AOR = 3.68, 95% CI = 1.73-7.82) more likely to occur among overweight/obese patients compared to their counterparts with normal BMI (Table 3).

## Discussion

Tight glycemic control is an important therapeutic goal for prevention of complication in patients with T2DM [20] and behavioral factors are the most important targets to achieve the set of diabetic goal. The American Diabetes Association's Standards of Medical Care focuses on diet, physical exercise and other life modifications for glycemic control [27].

However, in this study, 45.2% and 65.5% of diabetic patient were non-adherent to fruit and vegetable intake and physical exercise recommendations respectively. This study also

**Table 3. Behavioral determinants of glycemic control among type 2 diabetic patients attending chronic OPD clinic in WUNEMMTH, Hosanna, Central Ethiopia, 2024 (n = 226).**

| Variables | Classification | | | | P-value | COR | 95% C.I. | P-value | AOR | 95% C.I. |
|---|---|---|---|---|---|---|---|---|---|---|
| | | Cases | Controls | Total | | | | | | |
| Adherence to fruit and vegetables intake | Yes (Ref) | 43 | 81 | 124 | | 1 | | | 1 | |
| | No | 70 | 32 | 102 | <0.001 | 4.12 | 2.36–7.20 | <0.001 | 3.38 | 1.73–6.60 |
| Adherence to fiber intake | Yes | 37 | 74 | 111 | | 1 | | | | |
| | No | 76 | 39 | 115 | <0.001 | 3.90 | 2.24–6.77 | | | |
| Adherent to physical exercise | Yes (Ref) | 21 | 57 | 78 | | 1 | | | 1 | |
| | No | 92 | 56 | 148 | <0.001 | 4.46 | 2.45–8.13 | <0.001 | 4.94 | 2.38–10.27 |
| Cigarette smoking status | Non-smoker | 92 | 99 | 191 | | 1 | | | | |
| | Ex-smoker | 21 | 14 | 35 | 0.20 | 0.62 | 0.30–1.29 | | | |
| Compliance to medications | Yes | 72 | 98 | 170 | | 1 | | | | |
| | No | 41 | 15 | 56 | <0.001 | 3.72 | 1.91–7.23 | | | |
| Diabetic Self-efficacy | Poor (Ref) | 83 | 31 | 114 | <0.001 | 7.32 | 4.07–13.17 | <0.001 | 5.51 | 2.85–10.66 |
| | Good | 30 | 82 | 112 | | 1 | | | 1 | |
| BMI (kg/m2) | ≥25 | 45 | 20 | 65 | <0.001 | 3.08 | 1.67–5.68 | <0.001 | 3.68 | 1.73–7.82 |
| | <25 (Ref) | 68 | 93 | 161 | | 1 | | | 1 | |

COR = Crude Odds Ratio, AOR = Adjusted Odds Ratio, Ref = Reference, C. I = Confidence interval.

demonstrated that 50.4% of study participants had poor diabetic self-efficacy and 28.8% of the participants had high BMI.

The current study identified that status of adherence to fruit and vegetable intake, physical exercise, diabetic self-efficacy and BMI were statistically significant determinants for the glycemic control among T2DM patients.

In this study, the risk of poor glycemic control was about 3.4 times more likely to occur among non-adherents to fruit and vegetable recommendations. This is in line with studies conducted in various parts of the world. A study in Addis Ababa mentioned that poor compliance to dietary recommendation was associated with poor glycemic control [17]. A quasi-experimental study in North Shoa, Ethiopia, demonstrated that a dietary intervention had improved glycemic control through change in patients' behaviors towards adherence to healthy diet [28]. This is also consistent with a systematic review and meta-analysis which reported that adherence to dietary recommendation found to be the most significant predictor of glycemic control in T2DM patients [29]. Additionally, randomized control trials demonstrated that fruit and vegetable intake were major contributors to improved glycemic control among patients with T2DM [30,31]. The possible explanation is that fruits and vegetables are vital sources of functional foods, which are low in calories and rich in dietary fibers and phytochemicals like polyphenols. Dietary polyphenols, naturally abundant in fruits and vegetables, provide potential benefits for glycemic control in regulating plasma glucose levels [32].

In this study, the risk of poor glycemic control was about 5 times more likely to occur among T2DM patient who poorly adhere to physical exercise. This is supported by other studies. Certain studies in in Ethiopia mentioned that insufficient physical activity is an independent predictors of poor glycemic control among T2DM patients [20,33,34]. This might be due to increase in glucose uptake by the working muscle as a result of increase in the blood flow and eventually increase in the number of insulin receptors, which finally results in increasing insulin sensitivity [20].

This study also identified that diabetic self-efficacy was directly associated with glycemic control. The likelihood of poor glycemic control was 5.5 times more common to occur among patients who had poor self-efficacy towards diabetic management. This is due to evidences that perceived self-efficacy is a reliable variable in predicting healthy behavioral initiation for self-management to determine health outcomes among patients with chronic health conditions [26,29,30]. Patients with diabetes having higher self-efficacy were significantly more likely to have good glycemic control than their counterparts [35]. The likely explanation is that self-efficacy is a mediator between knowledge and healthy lifestyles, which poses direct impact on meeting one's goals in patients with a chronic disease. Self-efficacy, through adherence to healthy behaviors, is an important construct which is strongly related to glycemic control among T2DM patients [29,36–39].

Patient with high BMI were about 3.7 times more likely to go through poor glycemic control compared to their counterparts. This is also in agreement with other findings which reported that the risk of not achieving glycemic control in individuals with T2DM patients is increased with higher BMI [25,40,41]. This is further supported by studies which showed that lifestyle interventions targeting weight loss improves the overall glycemic control and lessen the need for glucose-lowering medications among T2DM [27,40]. The likely explanation for this relation could be indicated by the evidences in which obesity increases the secretion of insulin resistant non esterified fatty acids (NEFAs) from adipose tissue and consequently results in decreased insulin sensitivity, which in turn is a risk factors of T2DM [20,40]. However, a prospective study in china reported inconsistent result that baseline BMI had no effect on glycemic control [42].

## Conclusion and recommendations

Self-efficacy and modifiable lifestyles were significant determinants of glycemic control among T2DM patients. The findings may help in the customization of interventions for glycemic control among T2DM patients and aid health care providers in counseling patients regarding healthy behavior. Thus, targeting lifestyle is the most feasible approach for glycemic control in patients with T2DM.

Along with pharmacotherapy, diabetic self-care behaviors focusing on self-efficacy should be an integral part of the management. Health sectors should provide continuous health education emphasizing on lifestyle modification and adherence to fruit and vegetable consumption, physical activity, self-efficacy and control to body weight. The importance of intervention study is highlighted.

## Strength and limitations

This is a case control design that would show strong evidences for glycemic control particularly which aimed to identify modifiable determinants of glycemic control. To the best of our knowledge, this is the first study to reveal the role of self-efficacy in glycemic control. Despite such strengths, we recognize as a limitation that certain variables may have effect modification over the others. Furthermore, this study used FBG instead of HbA1c, which is a better parameter to evaluate status of glycemic control. Additionally, behavioral factors were measured by self-report questionnaires, therefore we could not have ruled out social desirability bias.

## Supporting information

**S1 Fig. Conceptual framework for behavioral determinants of glycemic control among patients with T2DM in Hosanna town, Central Ethiopia, 2024.**
(TIF)

**S2 File. Data collection tool.**
(DOCX)

**S3 Data Set. SPSS Data set**.
(SAV)

**S4 Fig. Adherence to healthy dietary pattern among type 2 diabetic patients in WUNEM-MTH, Hosanna, Central Ethiopia, 2024 (n = 226).**
(TIF)

## Acknowledgments

We would like to acknowledge Institutional Review Board of Hosanna Health Science College for reviewing and approving of this study. It is our sincere pleasure to acknowledge all who took part in this study specifically, study participants, research assistants and supervisors for their contribution towards this work.

All authors contributed significantly and gave the final approval for the paper to be published; agreed to be accountable for all impacts of the work.

## Author contributions

**Conceptualization:** Belay Bancha, Begidu Bashe.

**Data curation:** Belay Bancha.

**Formal analysis:** Belay Bancha.

**Methodology:** Belay Bancha, Begidu Bashe.

**Software:** Belay Bancha, Begidu Bashe.

**Supervision:** Belay Bancha, Begidu Bashe, Hana Seyfu, Abraham Samuel.

**Validation:** Belay Bancha, Begidu Bashe, Hana Seyfu, Abraham Samuel.

**Writing – original draft:** Belay Bancha.

**Writing – review & editing:** Belay Bancha, Hana Seyfu, Abraham Samuel.

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
