## [Decision Letter · Decision Letter 0]

25 Oct 2024

PONE-D-24-40268Behavioral determinants for glycemic control among type 2 diabetic patients in Hosanna town; institution based unmatched case control studyPLOS ONE

Dear Dr.  Bancha,

Thank you for submitting your manuscript to PLOS ONE. After careful consideration, we feel that it has merit but does not fully meet PLOS ONE’s publication criteria as it currently stands. Therefore, we invite you to submit a revised version of the manuscript that addresses the points raised during the review process.

**ACADEMIC EDITOR: ** :

**Introduction**Line 59. The burden is very significant in resource-limited settings. Rewrite it**Methodology **The authors include only three months and above. Does this mean that patient behavioral change needs at least three months to occur?The authors include AS A LAB. Reference FBS. Why Hgba1c, since it is more useful for glucose monitoring in the last three to four months.**Operational definition**Adherence to medication is defined as patients taking all his/her anti-diabetic medication in the last seven days. is it the standard definition? The patient may be non adherent before seven days but he/she might be adherent for the next seven days.How do the authors define self-efficacy in diabetic control

We look forward to receiving your revised manuscript.

Kind regards,

Bedilu Linger Endalifer

Academic Editor

PLOS ONE

Journal Requirements:

1. When submitting your revision, we need you to address these additional requirements. Please ensure that your manuscript meets PLOS ONE's style requirements, including those for file naming. The PLOS ONE style templates can be found at https://journals.plos.org/plosone/s/file?id=wjVg/PLOSOne_formatting_sample_main_body.pdf and https://journals.plos.org/plosone/s/file?id=ba62/PLOSOne_formatting_sample_title_authors_affiliations.pdf 2. When completing the data availability statement of the submission form, you indicated that you will make your data available on acceptance. We strongly recommend all authors decide on a data sharing plan before acceptance, as the process can be lengthy and hold up publication timelines. Please note that, though access restrictions are acceptable now, your entire data will need to be made freely accessible if your manuscript is accepted for publication. This policy applies to all data except where public deposition would breach compliance with the protocol approved by your research ethics board. If you are unable to adhere to our open data policy, please kindly revise your statement to explain your reasoning and we will seek the editor's input on an exemption. Please be assured that, once you have provided your new statement, the assessment of your exemption will not hold up the peer review process. 3. Your ethics statement should only appear in the Methods section of your manuscript. If your ethics statement is written in any section besides the Methods, please move it to the Methods section and delete it from any other section. Please ensure that your ethics statement is included in your manuscript, as the ethics statement entered into the online submission form will not be published alongside your manuscript. 4. Please upload a copy of Figures 1 and 2, to which you refer in your text on pages 4 and 12. If the figure is no longer to be included as part of the submission please remove all reference to it within the text.

Reviewers' comments:

Reviewer's Responses to Questions

**Comments to the Author**

1. Is the manuscript technically sound, and do the data support the conclusions?

Reviewer #1: Yes

Reviewer #2: Yes

2. Has the statistical analysis been performed appropriately and rigorously? 

Reviewer #1: I Don't Know

Reviewer #2: Yes

3. Have the authors made all data underlying the findings in their manuscript fully available?

Reviewer #1: Yes

Reviewer #2: Yes

4. Is the manuscript presented in an intelligible fashion and written in standard English?

Reviewer #1: Yes

Reviewer #2: Yes

5. Review Comments to the Author

Reviewer #1: - How is the complex relationship shown in Fig. 1 represented in your model? Did you include BMI/diet and BMI/exercise as interaction terms? These will likely share variance and should therefore be included separately and as interaction. In general, there should be more information on the model specification.

- line 229: Is it really "p-value of <0.25"? If you set a p-threshold of <0.05 as you stated, you should report only those results as significant that are below that (and report the exact p-values).

- line 230: "After adjusting for others" => for what exactly?

- The Discussion speaks of diabetic patients overall, but it would be more in line with the rest of the paper to discuss differences (and similarities) between those with poor and good glycaemic control. This would also tie in this paragraph with the one following.

- I recommend not using 3D bar charts as they visually distort the height proportion without adding any information. Use 2D bar charts instead.

- There are some typos and grammatical errors (often with regard to the use of singular/plural).

Reviewer #2: 1. Has this clinical trial been registered on the registration platform?

2. Blood glucose monitoring is also part of the usual care of type 2 diabetes, and whether the data you collect can conduct the analysis of this habit?

6. PLOS authors have the option to publish the peer review history of their article (what does this mean? ). If published, this will include your full peer review and any attached files.

**Do you want your identity to be public for this peer review?** For information about this choice, including consent withdrawal, please see our Privacy Policy .

Reviewer #1: No

Reviewer #2: No

---

## [Author Response · Author response to Decision Letter 0]

1 Nov 2024

Response to Reviewers

Dear editor/reviewers,

We acknowledge the concerns raised by our dear editors and reviewers, and we learnt extraordinary lesson during the revision of our manuscript. The following are point-by-point responses to the concerns.

1. Line 59: Re-written

2. Three months as a reference: We included patients under follow up at least for three months. This is to measure the outcome variable from FBS mean score. In settings where HgbA1c is not feasible for follow-up care, mean FBS used to evaluate the status of glycemic control. We cited this in our case and control identification section

3. Use of FBS over HgbA1c: The hospital does not have the glycated hemoglobin (HbA1c) test for monitoring the glycemic status; instead, FBS is in use as a part of standard of care. We used this data from patients’ record for this study after guaranteeing due ethical clearance and consequently permission.

4. Operational definition

a. Adherence: Adherence was measured from last seven days status to minimize the risk of recall bias, and this definition was in line with others and we recognized the previous evidences as a reference (cited).

b. Self-efficacy: it was based on 15 items with 5-point Likert scale and this was cited.

5. We made an extensive revision in the current version to comply with PLOS ONE's style requirements.

6. We sincerely regret any technical error that may have occurred in the submission form with regard to data sharing. Actually we said, “All relevant data are within the paper and its supporting information files” under the data sharing sections. The constructed PDF which we downloaded during submission likewise shows this. We shared the data set from our initial submission, and freely shared it in this submission. No restrictions were made on dataset

7. The Ethical statement only appears as a subheading under the Methods section.

8. In response to our esteemed editors, we revised figure 1 and uploaded as a supporting information.

9. We have reviewed our references, and did not come up with retracted papers. Very few of our references are WHO guidelines, Demographic and health survey reports of Ethiopia.

10. The supporting file, S4 SPSS data set.sav (In the current version S4_Data set.sav) is based on IBM SPSS version 23 software. We have re-checked it and faced no difficulty. Even we tried the file from the built pdf link, which was downloaded during submission, it works well. Additionally we shared it again.

Reviewer #1_Fig 1. We appreciate our esteemed reviewers’ comment on the issue. However, we recognize an issue as a limitation.

Reviewer #1_Fig 2. Thanks, it was shifted to 2D bar chart.

Reviewer #1_Line 229, p value 0.25 We have described this under the data analysis section of the methodology. Bivariate analysis was done to identify potential variables for multivariable logistic regression. Variables with p-value of <0.25 in bivariate analysis were entered into multivariate analysis. In adjusted model, the level of statistical significance was declared at a p-value of <0.05. Despite that an SPSS output gives us p value of 0, such figures are not real 0. P may approach to 0, but never be 0 in reality. In such cases we used p<0.001.

Reviewer #1_Line 230 The adjustment (multivariable logistic regression analysis) used all variables, which we identified in bivariate analysis. Theses variables were listed in the preceding paragraph. Additionally, we have paraphrased the statement in the revised version.

Reviewer #1_Discussion Concerning issues related to discussion, we were focusing on our objective, behavioral determinant for glycemic control. However if a given factor is either inversely or directly associated with poor glycemic control, we feel that it is also associated with good glycemic control.

Reviewer 2_ “Has this clinical trial been registered on the registration platform?” This is an observational study, which we believe that does not require registration. It has gone through necessary ethical review process. We have submitted the ethical clearance and all relevant files pertaining the paper.

Reviewer 2_ “Blood glucose monitoring issue: We have described this in under the section of “Identification of cases and controls” and “Data collection tools and procedures”.

Sincerely thanks for consideration

---

## [Decision Letter · Decision Letter 1]

13 Nov 2024

Behavioral determinants for glycemic control among type 2 diabetic patients in Hosanna town; institution based unmatched case control study

PONE-D-24-40268R1

Dear Dr. Belay Bancha

We’re pleased to inform you that your manuscript has been judged scientifically suitable for publication and will be formally accepted for publication once it meets all outstanding technical requirements.

Kind regards,

Bedilu Linger Endalifer

Academic Editor

PLOS ONE

Additional Editor Comments (optional):

All my concerns are addressed

Reviewers' comments:

Reviewer's Responses to Questions

**Comments to the Author**

1. If the authors have adequately addressed your comments raised in a previous round of review and you feel that this manuscript is now acceptable for publication, you may indicate that here to bypass the “Comments to the Author” section, enter your conflict of interest statement in the “Confidential to Editor” section, and submit your "Accept" recommendation.

Reviewer #1: All comments have been addressed

2. Is the manuscript technically sound, and do the data support the conclusions?

Reviewer #1: Yes

3. Has the statistical analysis been performed appropriately and rigorously? 

Reviewer #1: Yes

4. Have the authors made all data underlying the findings in their manuscript fully available?

Reviewer #1: Yes

5. Is the manuscript presented in an intelligible fashion and written in standard English?

Reviewer #1: Yes

6. Review Comments to the Author

Reviewer #1: Thanks for addressing my comments!

Very minor:

Page 9, line 184: "Variables" is still in upper-case despite "The" being added in front.

7. PLOS authors have the option to publish the peer review history of their article (what does this mean? ). If published, this will include your full peer review and any attached files.

**Do you want your identity to be public for this peer review?** For information about this choice, including consent withdrawal, please see our Privacy Policy .

Reviewer #1: No

---

## [Editor Report · Acceptance letter]

PONE-D-24-40268R1

PLOS ONE

Dear Dr. Bancha,

I'm pleased to inform you that your manuscript has been deemed suitable for publication in PLOS ONE. Congratulations! Your manuscript is now being handed over to our production team.

Kind regards,

on behalf of

Dr. Bedilu Linger Endalifer

Academic Editor

PLOS ONE